# Peri-Operative Management of Patients Undergoing Fenestrated-Branched Endovascular Repair for Juxtarenal, Pararenal and Thoracoabdominal Aortic Aneurysms: Preventing, Recognizing and Treating Complications to Improve Clinical Outcomes

**DOI:** 10.3390/jpm12071018

**Published:** 2022-06-21

**Authors:** Andrea Xodo, Mario D’Oria, Bernardo Mendes, Luca Bertoglio, Kevin Mani, Mauro Gargiulo, Jacob Budtz-Lilly, Michele Antonello, Gian Franco Veraldi, Fabio Pilon, Domenico Milite, Cristiano Calvagna, Filippo Griselli, Jacopo Taglialavoro, Silvia Bassini, Anders Wanhainen, David Lindstrom, Enrico Gallitto, Luca Mezzetto, Davide Mastrorilli, Sandro Lepidi, Randall DeMartino

**Affiliations:** 1Vascular and Endovascular Surgery Unit, “San Bortolo” Hospital, AULSS8 Berica, 36100 Vicenza, Italy; andreaxodo@me.com (A.X.); fabio.pilon@aulss8.veneto.it (F.P.); domenico.milite@aulss8.veneto.it (D.M.); 2Cardiovascular Department, Division of Vascular and Endovascular Surgery, Trieste University Hospital ASUGI, 34149 Trieste, Italy; clavagnacristiano@gmail.com (C.C.); fgriselli@gmail.com (F.G.); jacopo.taglialavoro@gmail.com (J.T.); sbassini@hotmail.it (S.B.); slepidi@units.it (S.L.); 3Gonda Vascular Center, Division of Vascular and Endovascular Surgery, Mayo Clinic, Rochester, NY 55902, USA; mendes.bernardo@mayo.edu (B.M.); demartino.randall@mayo.edu (R.D.); 4Division of Vascular Surgery, IRCCS San Raffaele Scientific Institute, “Vita-Salute” San Raffaele University, 58-20132 Milan, Italy; bertoglio.luca@hsr.it; 5Section of Vascular Surgery, Department of Surgical Sciences, University of Uppsala, 75236 Uppsala, Sweden; kevin.mani@surgsci.uu.se (K.M.); anders.wanhainen@surgsci.uu.se (A.W.); david.lindstrom@akademiska.se (D.L.); 6Vascular Surgery, IRCCS-University Hospital Policlinico S. Orsola, DIMES-University of Bologna, 40138 Bologna, Italy; mauro.gargiulo2@unibo.it (M.G.); enrico.gallitto@gmail.com (E.G.); 7Department of Cardiovascular Surgery, Division of Vascular Surgery, Aarhus University Hospital, 161-8200 Aarhus, Denmark; jacoblilly@me.com; 8Vascular and Endovascular Surgery, University Hospital of Padova, DSCTV-University of Padova, 35128 Padova, Italy; michele.antonello.1@unipd.it; 9Unit of Vascular Surgery, Integrated University Hospital of Verona, 37126 Verona, Italy; gianfranco.veraldi@aovr.veneto.it (G.F.V.); luca.mezzetto@aovr.veneto.it (L.M.); mastrodvd87@gmail.com (D.M.)

**Keywords:** aortic disease, aortic aneurysm, fenestrated-branched endovascular repair, outcomes, complications, review

## Abstract

The advent and refinement of complex endovascular techniques in the last two decades has revolutionized the field of vascular surgery. This has allowed an effective minimally invasive treatment of extensive disease involving the pararenal and the thoracoabdominal aorta. Fenestrated-branched EVAR (F/BEVAR) now represents a feasible technical solution to address these complex diseases, moving the proximal sealing zone above the renal-visceral vessels take-off and preserving their patency. The aim of this paper was to provide a narrative review on the peri-operative management of patients undergoing F/BEVAR procedures for juxtarenal abdominal aortic aneurysm (JAAA), pararenal abdominal aortic aneurysm (PRAA) or thoracoabdominal aortic aneurism (TAAA). It will focus on how to prevent, diagnose, and manage the complications ensuing from these complex interventions, in order to improve clinical outcomes. Indeed, F/BEVAR remains a technically, physiologically, and mentally demanding procedure. Intraoperative adverse events often require prolonged or additional procedures and complications may significantly impact a patient’s quality of life, health status, and overall cost of care. The presence of standardized preoperative, perioperative, and postoperative pathways of care, together with surgeons and teams with significant experience in aortic surgery, should be considered as crucial points to improve clinical outcomes. Aggressive prevention, prompt diagnosis and timely rescue of any major adverse events following the procedure remain paramount clinical needs.

## 1. Introduction

The refinement of complex endovascular techniques in the last two decades has revolutionized the field of vascular and endovascular surgery, thereby allowing safe and effective minimally invasive treatment of aneurismatic disease involving the juxtarenal, pararenal and thoracoabdominal aorta. Endovascular aneurysm repair (EVAR) represents the most frequently used treatment in patients with infrarenal abdominal aortic aneurysms (AAA), who present with suitable anatomy and high physiological and surgical risk for open surgical repair (OSR) [1]. EVAR procedures, associated with adjunctive tools such as parallel grafts (PG) or endostapling systems, are currently used to treat juxtarenal AAA (JAAA) by “off-the-shelf” solutions, especially in case of urgency or emergency conditions or with patients considered “unfit” for OSR [2,3,4,5,6,7,8]. Nevertheless, fenestrated-branched EVAR (F/BEVAR) represents a feasible and widely adopted technique to treat a wide range of conditions including JAAA, pararenal AAA (PRAA) and thoracoabdominal aortic aneurysms (TAAA), moving the proximal sealing zone above the renal-visceral vessels take-off while preserving their patency [9,10,11]. Indeed, it is well known that achieving a seal in a morphologically hostile aortic neck will make the repair more prone to lower durability over time, while the parallel graft technique is only recommended as a bailout alternative. Therefore, fenestrated-branched endografts (whether physician-modified, off-the-shelf, or custom-made commercially manufactured) would represent the best available endovascular treatment option for complex aortic diseases (Table 1). At the same time, F/BEVAR procedures are technically demanding operations that may be associated with severe complications and greatly impact patients’ health. In order to reduce complications and recognize or treat them early, different tools and skills are needed, and this starts from careful preoperative planning with dedicated surgical and anesthetic management pathways during the intraoperative and early postoperative phases. This requires significant experience best, achieved at dedicated centers for the treatment of complex aortic disease. The aim of this paper is to analyze and review, from an experts’ opinion, the perioperative management of patients undergoing F/BEVAR procedures for complex abdominal (JAAA and PRAA) and thoracoabdominal aneurismatic aortic diseases. It will focus on how to prevent, diagnose, and manage the complications ensuing from such complex interventions, in order to improve clinical outcomes.

## 2. Pre-Operative Evaluation and Patient Selection

Endovascular therapies have gained popularity, mainly because of their reduced invasiveness and lower incidence of major systemic complications, especially in high-risk surgical candidates [12,13]. Despite this, all endovascular procedures, in particular for the treatment of extensive aortic disease, should be preceded by a thorough evaluation of all major organ systems (including cardiac, pulmonary and renal function), to outline the general health and physiological reserve of the patient. Indeed, most patients with aortic aneurysms may present with concomitant central and peripheral atherosclerotic lesions owing to the similar risk factors, such as hypertension, smoking and aging, amongst others [14]. All these comorbidities may increase the surgical risk and may negatively impact long-term survival, which has been well documented for OSR. In a large study by Shepens and colleagues who analyzed 500 open TAAA repairs, the identified risk factors for late death were depressed ventricular function, increased age, postoperative dialysis needed and neurological deficit [15]. In that sense, it should be borne in mind that, in the contemporary endovascular era, a patient considered “unfit” for open surgical TAAA repair may not be necessarily considered “fit” for F/BEVAR. In fact, very often these patients may have different notable pre-operative comorbidities or critical clinical/anatomical issues that mandate consideration (advanced age, renal diseases, recent or previous acute myocardial infarction treated by heart surgery or percutaneous coronary interventions, hostile vascular accesses, or shaggy aortic morphology amongst others). In this regard, the EVAR-2 trial clearly showed how proceeding with elective EVAR in patients at high physiologic risk may reduce aneurysm-related mortality, but not improve overall survival [16]. Therefore, the decision as to whether a patient shall undergo an elective operation must weigh the expected benefits of a prophylactic operation (i.e., to prevent aneurysm rupture) with the perceived risk of the operation itself and of life expectancy of the patient. Unlike for AAA, there are no validated specific protocols or risk prediction models for patient selection of F/BEVAR [17]. However, routine cardiology consultations before the operation, including possible cardiac stress testing and coronary angiography for patients identified as “high-risk”, may reduce the rate of cardiac complications [18]. Specifically, patients unable to perform four metabolic equivalents (METS) of activity (ability to climb a flight a stairs without dyspnea) reliably should undergo extensive cardiovascular evaluation [19]. Cardiac evaluation should preferably be performed non-invasively, for example with an echocardiogram, cardiac stress test, myocardial scintigraphy or coronary computed tomography angiography (CTA). If percutaneous coronary intervention is needed to reduce cardiac risk, dual antiplatelet therapy (DAPT) may delay surgery. Lumbar puncture for cerebrospinal fluid drainage (CSFD) is generally not recommended while on DAPT, due to the risk of spinal hematoma [20]. The prevalence of internal carotid artery stenosis is high among patients with AAA (8.8% in the SMART study that analyzed 2.274 patients with AAA). For this reason, a routine screening for asymptomatic carotid stenosis by Doppler ultrasonography before aortic aneurysm repair is usually carried out in daily clinical practice. Preoperative imaging studies must include thoraco-abdominal computed tomography angiography (CTA) from the neck to the groin, with multi-slice cuts of 1–3 mm. This is needed for planning and sizing of the endograft, as well as for the study of critical anatomic issues, such as angulation and diameter of the aorta and of the iliac vessels, calcifications or stenosis of the main aortic branches, suitability of the aortic arch in case upper extremity access, significant aortic mural thrombus, number of intercostal or lumbar arteries. Although F/BEVAR may be conceptually considered similar approaches, several anatomical challenges may be anticipated on pre-operative CTA that may hinder long-term durability of the repair, such as the presence of significant gaps in case of fenestrations or the use of long bridging stents in case of branches (Figure 1).

FEVAR is usually considered in cases with narrower aortic lumens (<25 mm) and target vessels (TV) perpendicular to the aorta or tilted upwards. Conversely, BEVAR is usually preferred in patients with larger aortic lumens (>25 mm) and downward angled TV. Planning is usually easier in the case of an “off-the-shelf” multibranched device, making this approach more suitable for urgent or emergency settings. However, those cases in which the aortic morphology is not ideal for an off-the-shelf design will mandate custom-made endografts that will fit the patient’s anatomy (Figure 2). These follow strict planning and sizing criteria, thereby requiring more prolonged manufacturing and delivering times [21].

Pre-operative CTA may also be used to assess the size and quality of the psoas muscle, as the presence of sarcopenia has been linked to poorer short-term and mid-term outcomes in patients undergoing F-BEVAR [22]. However, sarcopenia alone is not sufficient to predict surgical and clinical outcomes and should be used in the context of other factors to balance the perceived risk of aneurysm rupture and expected complexity of the interventional procedure. The importance of dedicated multidisciplinary teams for optimal assessment of patients’ fitness for such complex procedures may not be overemphasized enough. The use of frailty scoring tools by dedicated geriatric physicians may further complement the pre-operative evaluation of these patients, especially those of older age and/or with poor functional status.

## 3. Frequency and Impact of Post-Operative Complications after F/BEVAR and How They May Affect Outcomes and Costs Compared to EVAR

As compared with standard EVAR, F/BEVAR may be associated with higher complications and reinterventions rates, which is intuitively due to the complexity of disease and the need for renal-visceral vessels manipulation [23]. Locham and colleagues published a large study including 16,048 patients who underwent elective EVAR vs. 1641 patients who underwent elective FEVAR. The rates of any complications (9.6% vs. 11.3%, *p* = 0.03), renal injury (4.3% vs. 5.8%, *p* = 0.004) and neurologic injury (0.4% vs. 0.7%, *p* = 0.02) were all significantly higher in the FEVAR group compared with the EVAR group [24].

Indeed, the most frequent treatment-related complications that may arise following F/BEVAR interventions are represented by TV-related endoleaks (EL), which may occur in 1 out of 10 patients on CTA at discharge (Figure 3), although different adverse events may arise that can significantly impact patients’ health and quality of life [25,26]. TV instability was defined by a composite of any stent stenosis, separation, or type IC or type IIIC EL requiring reintervention and stent occlusion, aneurysm rupture or death due to TV-related complications [26]. Although it may seem likely that less complex FEVAR procedures will entail lower risks to the patients as compared with more extensive BEVAR interventions, the current evidence is mixed, as experienced operators may achieve comparable outcomes. Nonetheless, the length of aortic coverage and number of TV involved will likely remain as the main contributors of peri-operative adverse events also at centers of excellence and should, therefore, be included in the decision-making process [27]. Certainly, the team and surgeons’ experience are fundamental to preventing and recognizing early postoperative complications to ensure prompt and effective treatment. Furthermore, the rate of mortality and major complications after complex aortic surgery appears to be related to the surgical volume of the center, as shown by prior studies, mainly relating to the outcomes from major centers [28]. More than 154,000 patients underwent OSR and EVAR for intact AAA were analyzed by Scali and colleagues to identify an optimal center volume associated with the most significant mortality reduction after OSR; 65,745 underwent OSR and a significant inverse relationship between increasing center volume and lower peri-operative mortality after intact and ruptured OAR was evident (*p* < 0.001). An annual center volume of between 13 and 16 procedures per year (PY) was associated with the most significant mortality reduction after intact OAR (adjusted predicted mortality <13 PY 4.6% vs. 3.1% for centers with ≥13 PY) [29]. Data from a recent nationwide mandatory quality registry showed similar results for 694 patients, treated by FEVAR (539) and BEVAR (155). The perioperative mortality of complex EVAR was 9.1% in hospitals with a volume of <9 PY and 2.5% in hospitals with a volume of ≥13 PY (*p* = 0.008). The annual volume of ≥13 PY was associated with less perioperative mortality compared to hospitals with a volume of <9 PY [28]. Whether these outcomes can be extrapolated to low-volume operators to achieve similar outcomes is unclear. However, there is evidence that suggests a learning curve effect may exist for F/BEVAR interventions. Together with improved patient selection, these factors may substantially contribute to enhanced surgical outcomes [30]. Lately, the concept of failure to rescue (which is a composite endpoint defined as peri-operative death after experiencing at least one major complication) has emerged as an additional and more sensitive marker for team quality assessment in aortic interventions. Future research may also be directed toward better understanding of failure to rescue after complex endovascular aortic repair, as this remains relatively unaddressed in the available literature [31].

In addition to the clinical impact of complications, complex aortic endografting is usually more expensive compared with standard EVAR. In a European academic hospital, the costs of an EVAR procedure were EUR 12,090–13,956, while a FEVAR procedure costs EUR 34,807–36,695. In this experience, after the endograft costs, the costs of the hospitalization were the major contributors to the total cost of the procedures [32]. Although the cost of the stent-grafts may ultimately remain the main driver of expenditure, occurrence of post-procedural complications will increase the cost of care, and thus decrease the overall cost-effectiveness of treatment [33]. In fact, an extensive analysis of approximately 17,000 patients treated in the United States for aortic disease found that FEVAR procedures compared to EVAR interventions were associated with a significant increase in probability of renal and neurological damage. Furthermore, the total cost of FEVAR was significantly higher than standard EVAR and this appeared to be due to the additional cost of endovascular devices and to the increased complication rate related to FEVAR [24].

## 4. Ischemic vs. Hemorrhagic Complications: The Relevance and Balance of Antithrombotic Drugs

Owing to the extensive aortic coverage and involvement of critical renal-mesenteric vessels in the repair, management of antithrombotic therapy in F-BEVAR can be complex. Antithrombotic therapy requires a careful balance between the benefits of obtaining the optimal inhibition of the coagulation cascade to ensure TV patency versus the risks of serious bleeding events (including potential lethal complications, such as intracranial hemorrhage) [34]. Significant blood loss and blood product transfusion requirements have been shown to be associated with poorer clinical outcomes [35]. Patients scheduled for F/BEVAR may receive various antithrombotic drugs during their therapeutic path, starting from home drug therapy, moving to the intraoperative phase, and finally during long-term follow-up. Important factors impacting the use of anti-thrombotic and anti-coagulation therapy that are considered in the peri-operative setting of F-BEVAR include the potential need for cerebrospinal fluid drainage (CSFD), EL rates and visceral vessels and bridging stents’ (BS) patency. Unfortunately, there is a lack of robust clinical practice guidelines and the available literature is sparse. Therefore, the best available evidence will come from expert opinion, as it was recently reported in an international Delphi consensus of clinical practice. The optimal management of antithrombotic therapy was investigated amongst a large group of vascular surgeons with established experience with complex aortic interventions, with the aim to create recommendations on preoperative, intraoperative, and postoperative use of antithrombotic drugs in patients awaiting elective F/BEVAR. Based on expert opinions, very strong recommendations suggest that low dose aspirin should be considered in all patients planned for F/BEVAR, while dual antiplatelet therapy (DAPT) should be discontinued 7 days before the procedure if possible, in case of high risk of SCI [36,37]. Most experts also agree that intraprocedural unfractionated heparin at the beginning of a F/BEVAR procedure should be administered with a dosage of at least 70 UI/Kg and up to 100 UI/Kg, reaching an activated clotting time (ACT) of 200–250 s. In the post-operative phase, DAPT for at least one to six months may be considered, but possibly lifelong in patients with small and highly tortuous reno-visceral vessels, or when multiple bridging stents are used.

## 5. Intraoperative Endograft-Related and Target Vessels-Related Complications

The technical success of TAAA endovascular repair is related to the appropriate selection of an adequately sized endograft implanted in a safe and healthy proximal and distal landing zone (Figure 4).

This represents a crucial issue in guaranteeing durability over time and freedom from type 1A EL, in particular for Crawford type I or II TAAA. The type of arch also seems to influence the freedom from proximal endograft failure. In the case of type II or III aortic arches, the safest landing zone seems to be that of 30 mm, compared to 20–25 mm for type I arches [38].

Visceral vessel cannulation, wire insertion, and placement of bridging stents represent opportunities for potentially serious complications related to F/BEVAR, such as embolization or dissection. Once the TV has been cannulated, there is a risk of injury to the artery itself, with vessel perforation, or the end-organ it supplies, especially with renal arteries (Figure 5) [39].

In the case of FEVAR, more than for BEVAR, the correct positioning of the main body is fundamental for successful incorporation of the renal and splanchnic arteries. Perfect alignment will make it feasible to catheterize and stent the TV, while a twist of the main body may lead to misalignment with the TV, resulting in its occlusion and potentially leading to catastrophic consequences, unless promptly rescued during the intra-operative phase (Figure 6).

The use of diameter reducing ties allows for graft manipulation with partial deployment to mitigate this risk. A branched endograft allows a wider margin of tolerance for main body positioning compared to FEVAR, although the incorrect deployment of the endograft may still happen (Figure 7).

Regarding TV cannulation, the identification of any significant ostial stenosis is necessary, since this will make intraoperative maneuvers extremely demanding. Catheterization and bridging stent delivery are more complex in the setting of occlusive lesions in the visceral vessels. Ideally, when a FEVAR is planned, the endograft should be positioned as close as possible to the inner aortic wall in order to minimize the gap between the aortic graft and the TV, since this may result in short-term and long-term branch instability. The choice of bridging stent for the renal-mesenteric arteries are not standardized. There are not specifically designed BSs for F/BEVAR, but rather current stents are used in an “off label” manner for this purpose. To ensure effective and durable stability of the BS, ideally it should include a combination of different characteristics, such as consistent radial force, adequate flexibility, and a smooth transition at its distal edge to reduce the risk of intimal hyperplasia. Tenorio and colleagues analyzed 126 patients for an overall number of 335 renal-mesenteric arteries treated during BEVAR for TAAA, respectively, with self-expandable stent grafts (SESG) for 176 arteries and balloon-expandable stent grafts (BESG) for 159 arteries. TV instability occurred in 27 directional branches (8%), but SESG demonstrated higher primary patency, freedom from TV instability, freedom from type IC or type IIIC EL and freedom from TV-related reinterventions compared with BESG [26]. In addition, the length of the BS and their tortuosity index are important factors for good outcomes during BEVAR. A review of 32 patients with TAAA incorporating 123 arteries assessed the effect of the length and tortuosity of directional branches on the mid-term outcomes of branched endovascular aneurysm repair BEVAR for TAAA. The lowest branch instability rates were obtained with a branch total length of 60 to 100 mm, with a higher risk of branch related complications for a tortuosity index > 1.15. Furthermore, the renal arteries’ orientations affect the renal outcome of F/BEVAR, with worse outcomes for upward or downward/upward renal arteries’ orientation [40,41]. With regard to FEVAR, newer generations of BESG have shown promising early results, but future cross-comparisons of clinical series will be needed to assess their effectiveness and durability [42]. The technology available in the operating room will also contribute in several ways to the success of the surgical procedure. For example, the use of 3D fusion imaging is associated with a significant reduction in the amount of contrast media and radiation doses for complex endovascular aortic procedures, compared to standard biplanar roadmapping [43]. Newer technologies and techniques, including post-procedural assessment of BS quality with on-table cone-beam CT and intravascular ultrasound (IVUS) have also shown promise as effective tools for allowing prompt identification and resolution of technical issues that may be detrimental to outcomes [44,45,46,47]. IVUS after BS deployment is highly sensitive in the identification of morphological defects, allowing in-vivo sizing and a precise assessment of the distal sealing of the BS and detecting intraoperative issues missed by angiography [46,48].

## 6. Neurologic Complications: Stroke and Spinal Cord Ischemia

Although fenestrated-branched endografts are associated with decreased early mortality rates over OSR, F/BEVAR for TAAA treatment is still associated with a 0–33% risk of spinal cord ischemia (SCI) [49,50]. This risk can be reduced, but not completely eliminated with different adjuncts, such as preconditioning of the paraspinal collateral network, systemic hypothermia, distal aortic perfusion, and use of motor and somatosensory evoked potentials, as well as peri and postoperative drainage of cerebrospinal fluid (CSF) during and after aortic aneurysm repair. A retrospective study conducted by Spanos and colleagues on 243 patients treated by F/BEVAR for PRAA and TAAA showed a total incidence of SCI as high as 18%. In this experience, SCI was associated with reduced pre-operative renal function and the number of vertebral segments covered [51]. Prior AAA repair, prolonged hypotension, severe atherosclerosis of the thoracic aorta, left subclavian artery (LSA) or internal iliac artery (IIA) occlusion and extensive coverage of the thoracic aorta (>20 cm) by the endograft have all been associated with an increased incidence of SCI, in particular during thoracic endovascular repair (TEVAR) [52,53,54,55]. The staging of more complex procedures has increased in contemporary practice to increase the spinal cord tolerance and reduce the occurrence of SCI after extensive aortic endografting. A multistage approach to the endovascular treatment of TAAA by F/BEVAR is mainly based on the concept of collateral network remodeling after acute blood supply modifications (Table 2).

Staging can be performed using different strategies, such as minimally invasive segmental artery coil embolization (MIS^2^ACE) and temporary aneurysm sac perfusion (TASP) (Figure 8).

Chris Etz and colleagues described selective preventive catheter-guided embolization and occlusion of segmental arteries. This technique has demonstrated promising results in a limited experience and is currently under investigation in a prospective multicenter randomized trial [56,57]. The TASP concept may also be applied in different ways, including leaving an unbridged branch, performing sequential aortic coverage with stent-grafts, and delaying iliac limb cannulation in the bifurcated distal body. A recent study by Bertoglio and colleagues analyzed 240 patients treated for TAAA by F/BEVAR, where 43 patients had an impaired collateral network, 136 had a historical staging (previous OSR or endovascular procedure with intercostal/lumbar arteries ligation or coverage), and 157 received a staging procedure. The overall rate SCI is also negatively affected by impaired collateral network and bilateral iliac occlusive disease; for this reason, aggressive revascularization of the LSA (preferentially by means of a carotid-subclavian bypass or with custom-made fenestrated-branched proximal TEVAR component) and the IIAs (with preference given to the use of iliac branch devices, bilaterally if needed) should be pursued in order to increase perfusion to the spinal collateral network [58,59,60,61]. Early lower limb and pelvic reperfusion, with possible use of adjuncts such as pre-loaded guidewire systems, has also been shown to impact the onset and severity of SCI after F/BEVAR [62,63,64]. The potential of leaving distal lumbar arteries uncovered by designing a straight stent-graft that will land in otherwise healthy native infrarenal aorta (i.e., avoiding the placement of distal bifurcated EVAR body) has been recently suggested by some groups and will need further validation of its durability [65,66,67].

The use of lumbar spinal drains has historically been used to prevent SCI, but it is associated with high rates of complications that may exceed the benefits in the setting of patients at low risk for permanent neurologic injury. This has led some authors to suggest it be employed only as a rescue maneuver in case of SCI onset, whereby its prophylactic use must be restricted to a small subset of patients deemed at very high risk for post-operative paraplegia/paraparesis [68,69] In a cohort of 187 patients treated by F/BEVAR with catheter placement, 19 patients (10%) developed a complication related to this, 17 of these moderate or severe, with 4 episodes of spinal hematoma paraplegia or paraparesis. Complications of CSF drainage are, therefore, common, and its use should be weighed carefully. Tenorio and colleagues developed a protocol, used in patients with TAAA that required coverage of >5 cm above the celiac artery. This protocol included CSF drainage, neuromonitoring, lower extremity reperfusion, and selective temporary sac reperfusion. In 55% of the patients, neuromonitoring changes led to intraoperative maneuvers with an improvement of the neuromonitoring tracing in 90% of the patients. This resulted in an SCI rate of 4% (one without alteration of motor potentials) and 1% rate of permanent paraplegia [70]. Since BEVAR introduction, upper extremity access has been considered mandatory for cannulation and stenting of TV; however, the major complication of this access is a 2–4% risk of stroke, due to the crossing and maneuvering of sheaths through the aortic arch [71,72,73]. The recent adoption of steerable sheaths has enabled a totally transfemoral approach to BEVAR, which may reduce stroke rates for these procedures. Eilenberg et al. analyzed 152 consecutive patients who underwent BEVAR, 60 of whom were treated by a transfemoral approach (TFA) and 92 by an upper extremity approach (UEA). The technical success was higher for the TFA group (100% vs. 95%, *p* < 0.01), and perioperative strokes and transient ischemic attacks only occurred in the UEA group (8.6% vs. 0%, *p* = 0.022) [74]. However, concerns exist that a TFA approach may portend higher rates of SCI, owing to the need for longer ischemic times in the lower limbs bilaterally. Additionally, some anatomical conditions may render this approach unfeasible and will need to be assessed in the future. Overall, a stroke after endovascular aortic procedures remains multifactorial in nature, due to the landing zones in or near the aortic arch, coverage of the LSA, intraoperative embolization from the manipulation of endovascular devices, the use of upper extremity access, and presence of significant atherosclerotic debris. Therefore, an effective prevention strategy must rely on a multimodal approach and may not be limited to a single item. Aggressive revascularization of the LSA should be pursued in the elective setting, as its coverage may be related to higher rates of cerebrovascular accidents. In addition, coverage of the LSA impairs flow to the vertebral artery that impact the collateral network of the anterior spinal artery and can increase the risk of SCI after extensive aortic repair. Although carotid-subclavian bypass or transposition currently remains the gold standard to revascularize the LSA, newer endovascular alternatives (such as the use of fenestrated or branched endografts for the LSA) have been described recently. Their use is novel, and their actual durability still remains to be assessed [75,76,77,78,79,80].

## 7. Acute Kidney Injury and Other Ischemic Complications (e.g., Ischemia of the Bowel or Lower Extremity)

The revascularization of the renal and visceral arteries during complex aortic surgery entails arterial reconstruction and periods of inherent ischemia until the repair is completed; this can contribute to a variety of complications, whose severity may range from mild to life or organ threatening events with fatal consequences [81]. Prior studies have shown that the overall incidence of ischemic complications may be intrinsically higher in F/BEVAR procedures as compared with standard EVAR, owing to the higher complexity of such procedures [82]. The incidence of acute kidney injury (AKI) has been shown to be higher for patients with pre-existing chronic kidney disease (CKD), as defined by eGFR <60 mL/min/1.73 m^2^, compared to those patients with unimpaired renal function (12% vs. 37%, *p* < 0.001). However, this increased risk of AKI does not appear to correlate to higher peri-operative mortality, as compared with patients with unimpaired renal function [83,84]. AKI is multifactorial in nature and its prevention relies on a multimodal approach that includes minimization of device manipulation (especially in shaggy aorta), reduction in the iodinated contrast medium, peri-operative hydration with 0.9% saline, and withdrawal of nephrotoxic drugs. Ideally, newer technological advancements for intraoperative navigation during F/BEVAR, such as the CO_2_-based angiography or fiber optic real shape technology, may further contribute in the near future, since they may minimize these risk factors [85]. Bowel ischemic complications represent a rare (incidence of 1–3%) but highly morbid event if not corrected immediately [80]. If not amenable to endovascular salvage, laparotomy should be promptly performed, and retrograde extra-anatomic bypass (from the common iliac to the superior mesenteric artery (SMA) for example) or antegrade (aorto-mesenteric bypass if not included in the stent graft zone) offers an potentially expedient revascularization of the SMA [86]. Select TAAA (some extent I/V) can be treated without resorting to a fenestrated or branched endograft under selected circumstances, using TEVAR with PG techniques or a hybrid repair of the viscera. Coverage of the celiac trunk appears to be safe in terms of freedom from type 1 EL and aortic-related mortality, provided proper collateralization from the SMA is present [87,88]. However, ischemic foregut complications are possible. Thus, patient selection and a thorough preoperative determination of adequate collateral perfusion between the celiac trunk and SMA is mandatory if the former will not be incorporated in the repair [89].

The quality of femoral and iliac artery accesses is a crucial factor for any type of endovascular procedure, in particular for F/BEVAR. These procedures require the use of large introducer sheaths (20–24 Fr) and access site complications have been reported to occur in 8–16% of cases [90]. More extensive atherosclerotic disease and smaller access artery diameter present a higher risk for both bleeding complications and ischemia/reperfusion injuries. This is more commonly observed in female patients and poises special consideration to avoid such complications [91]. A clinical study of 231 patients undergoing F/BEVAR for TAAA or PRAA demonstrated a high technical success (93%) of totally percutaneous femoral access (TPFA) in preoperatively selected patients. Approximately 70% of patients were eligible for a TPFA, considering the position of the femoral bifurcation and the anatomical features of the femoral-iliac axis [92]. Therefore, a percutaneous approach to F/BEVAR currently represents the gold-standard for management; however, surgical cutdown should be considered in patients with unfavorable access anatomy. For severe iliac artery occlusive disease, a surgical or endovascular “conduit” may be required to guarantee safe access to the infrarenal aorta [93,94]. The presence of diseased ilio-femoral axes may greatly impact the technical demands of the operation or lead to lower extremity ischemia [95]. Newer-generation low-profile endografts may contribute to further reduction in the rate of lower limb complications, although their availability is limited by current stent technology [96].

## 8. Manufacturing Time-Related Adverse Events

Manufacturing time for custom-made endografts is a time-consuming process that usually requires 4 to 12 weeks for manufacturing and delivery. During the waiting time, there is an inherent risk of both aneurysm rupture and patient decline from other medical conditions (heart or renal failure episodes). This may lead to postponement or cancellation of the planned procedure, with a considerable expense in the case of custom-made devices [97]. Recent evidence has shown that this delay will ultimately lead to the cancellation of the surgical procedure up to 3% of the time [98]. A large diameter of the aneurysm (>70 mm) has been found to be associated with a higher risk for rupture during the waiting time, and in such cases, the use of off-the-shelf alternatives might be considered [99]. It is important to underline that the waiting time after the treatment decision can be differentiated into two distinct phases, namely the time required for the endograft to be manufactured and shipped, and the time needed for the operation to be scheduled after the endograft is available at the treating facility. Therefore, physicians may have the opportunity to reduce the overall waiting time only by scheduling the procedures as close as reasonably achievable to the delivery date. For the reasons mentioned above, it is also evident that all patients undergoing a customized procedure should have their concomitant underlying medical diseases and comorbidities closely managed in the interim until repair.

## 9. Gaps in Knowledge, Implications for Practice, and Future Directions of Research

Although their role in the management of patients with extensive aortic disease is currently well recognized, F/BEVAR interventions still involve a relatively low overall number of procedures in the world, performed at dedicated centers, compared to standard EVAR. As such, several topics related to F/BEVAR have yet to be fully explored. The long-term durability and choice of the BS during B/FEVAR is still uncertain. This includes the type of BS to be used for each TV type and the use of SESG versus BESG [21]. Further research is also required to better understand the complex geometrical factors that may contribute to TV-related outcomes after F/BEVAR [100]. Post-implantation syndrome has been extensively studied for patients undergoing EVAR, but there are few studies on this topic in patients treated by F/BEVAR. Given the greater number of implanted stent-grafts/materials, it is proposed that this is likely to be more widespread with unclear implications [101]. Whether the use of a percutaneous approach to the axillary artery will lead to improved clinical outcomes as compared with conventional brachial artery cutdown, or any significant differences exist with use of the right-side upper arm access vs. the left-side approach also remain areas of ongoing research efforts [102,103,104]. In addition, many of the perioperative factors discussed in the preceding chapters (such as procedural staging and risk for death during waiting time, optimal patient selection, type of BSs and use of ancillary materials, application of CSFD, or treatment for particular cases, such as post-dissection aneurysms, connective tissue diseases, or failures after prior infrarenal procedures) are only partially understood and represent unmet critical issues needing further study [105,106]. Lastly, while several papers discuss the conversion to OSR after failed EVAR [107,108,109], conversions after F/BEVAR are poorly described in the literature [110]. Therefore, it remains crucial to collect data in a prospective fashion, ideally with external validation of the data (e.g., by core-lab adjudication of relevant outcomes), with the incorporation of both patient-reported outcome measures of quality of life as well as cost metrics. Multicentric collaboration, such as the US Fenestrated-Branched Aortic Research Consortium, the Italian Multicenter Fenestrated and Branched Study Group or Dutch Surgical Aneurysm Audit, are ideally poised to provide these data [111,112]. Harmonization of the collected metrics in adherence to the reporting standards should be pursued to allow quality research into these needed areas.

## 10. Conclusions

Complex F/BEVAR for extensive aortic disease remains a technically and physiologically demanding procedure, and intra and postoperative complications may arise in up to 10–30% of cases [50,53]. These complications can impact the quality of life and health status of the patients and dramatically increase the cost of care. The utilization of standardized preoperative, perioperative, and postoperative pathways of care, together with high volume surgical teams, should be considered as critical points to improve clinical outcomes. Aggressive prevention, prompt diagnosis, and timely rescue of any major adverse events following the procedure remain paramount [107]. Further multicenter studies are needed to better address the many unsolved issues of this emerging and evolving technology in surgical care.

## Figures and Tables

**Figure 1 jpm-12-01018-f001:**
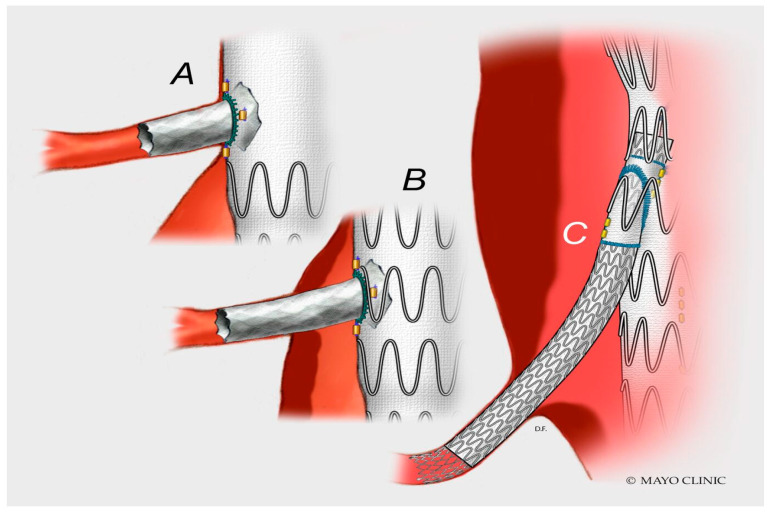
Technical solutions for bridging of target vessels during F/BEVAR. (**A**) Short balloon-expandable stent-graft for juxtarenal AAA treated with FEVAR; (**B**) long balloon-expandable stent-graft for suprarenal AAA treated with FEVAR (the gap between fenestration and inner aortic wall may lead to target vessel instability); (**C**) self-expanding stent-graft for TAAA treated with BEVAR and adjunctive distal relining with bare metal stent to accommodate smooth transition between edge of stent-graft and native artery in a tortuous segment.

**Figure 2 jpm-12-01018-f002:**
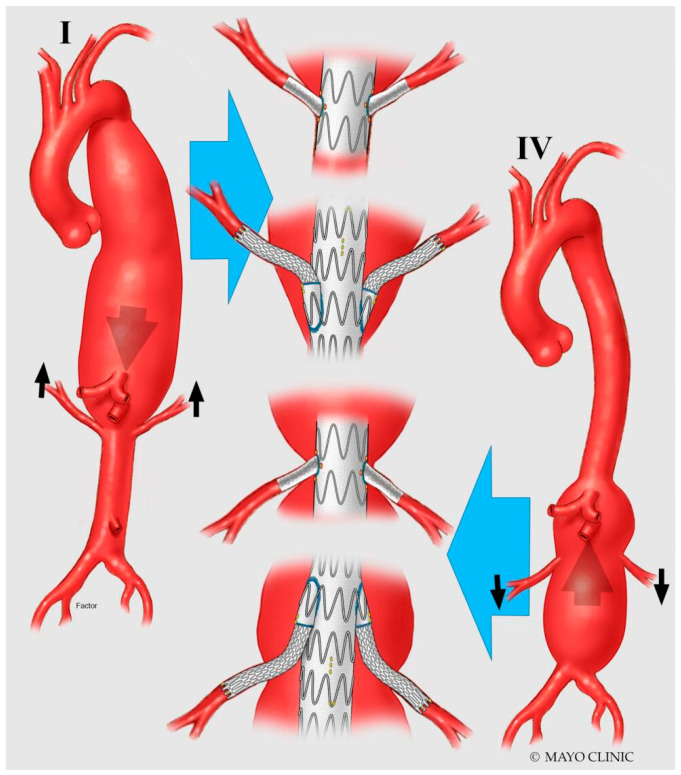
Technical solutions with different configurations (upward outer branches, inner branches, downward outer branches) for incorporation of renal arteries during BEVAR. I: complex AAA with upward orientation of renal arteries; IV: complex AAA with downward orientation of renal arteries.

**Figure 3 jpm-12-01018-f003:**
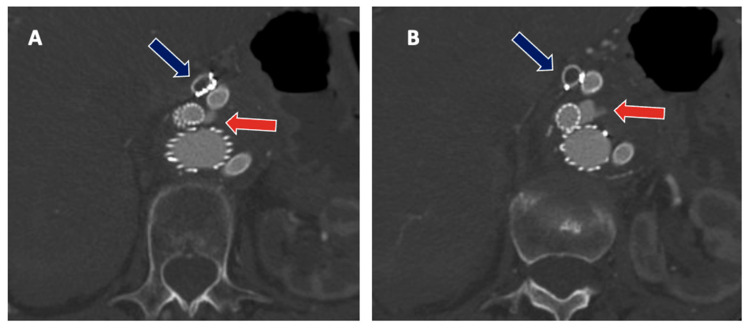
(**A**,**B**) Type IIIb endoleak from the right renal stent (red arrow), with sac enlargement and simultaneous asymptomatic thrombosis of the celiac trunk stent-graft (blue arrow).

**Figure 4 jpm-12-01018-f004:**
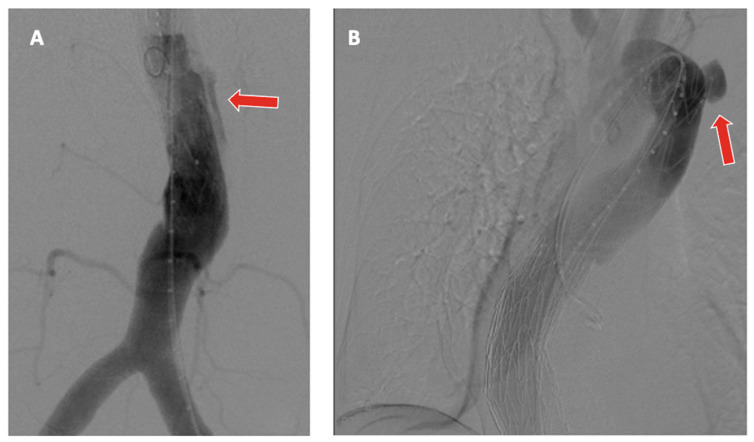
(**A**) Type 1B endoleak after BEVAR (red arrow). (**B**) Aortic rupture in zone 4 (Ishimaru’s classification) after TEVAR (red arrow) and BEVAR procedure to treat a large type III TAAA.

**Figure 5 jpm-12-01018-f005:**
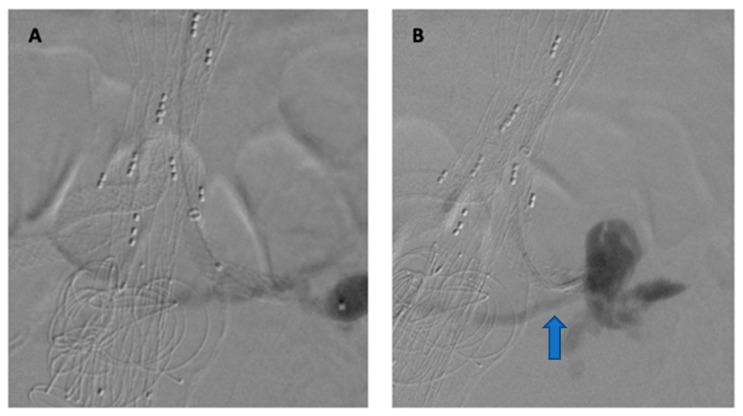
(**A**,**B**) Distal left renal artery rupture after bridging stent deployment during a TAAA repair, as observed on selective angiography (blue arrow).

**Figure 6 jpm-12-01018-f006:**
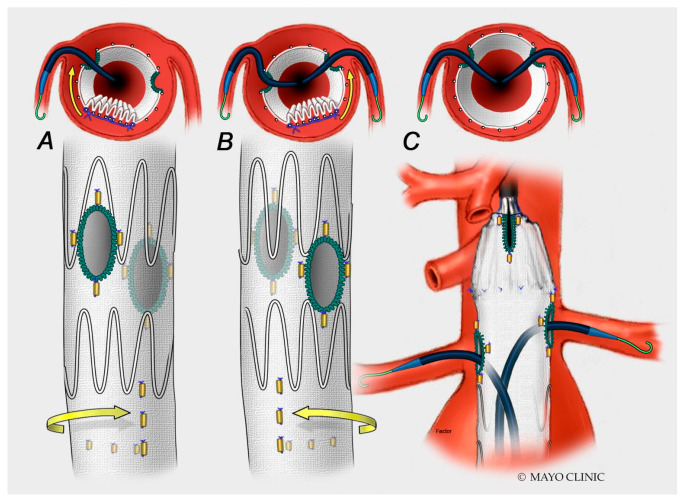
The presence of constraining wires on the back of the endograft allows for some degrees of rotation of the endograft (when partially deployed) in order to facilitate cannulation of the target vessels until the position is stabilized with the use of long introducers and the endograft may be completely deployed. (**A**) Counterclockwise and (**B**) clockwise rotation of the stent-graft to facilitate catheterization of target vessels; (**C**) sheaths inplace in the renal arteries before releasing the diameter-constraining wires on the stent-graft.

**Figure 7 jpm-12-01018-f007:**
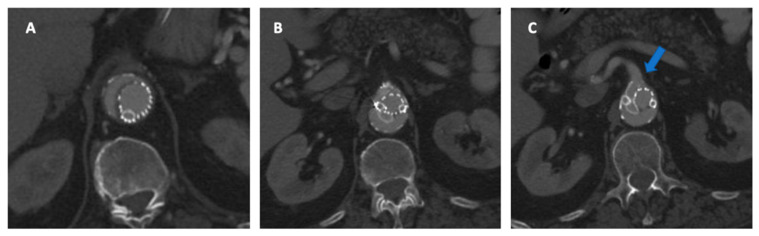
During this procedure, the branched endograft was positioned incorrectly (turned 180°). Note in (**A**,**B**) the posterior origin of the branches for celiac trunk and superior mesenteric artery, respectively. In (**C**), there is evidence of a type 1C endoleak due to an inadequate sealing zone in the superior mesenteric artery (blue arrow).

**Figure 8 jpm-12-01018-f008:**
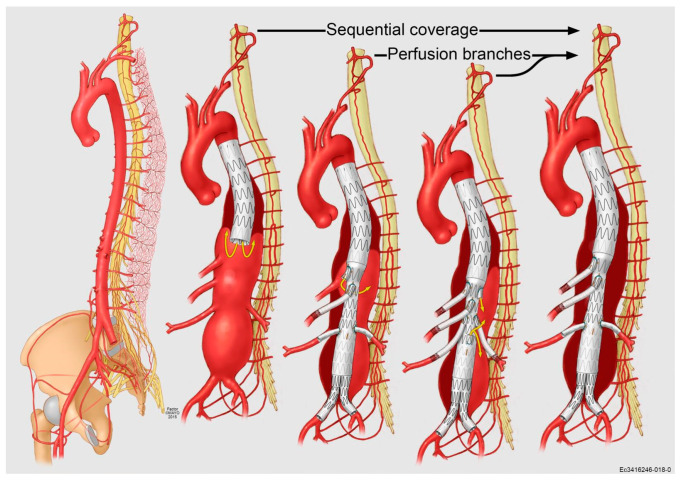
Sequential coverage of the aorta with continuous perfusion of the left subclavian and hypogastric arteries will permit safe and efficient development of the spinal cord collateral network, thereby allowing for reduction in the risk of spinal cord ischemia even after extensive endografting for TAAA. Yellow arrows indicate sources of continued aneurysmal sac perfusion to allow for staging of the procedure through continued perfusion of the spinal cord collateral network.

**Table 1 jpm-12-01018-t001:** Summary of EVAR and F/BEVAR indications, pros and cons.

	Standard EVAR	F/BEVAR (Custom-Made Device)	BEVAR (Off-The-Shelf Device)
Indications	Infrarenal AAA	JAAA/PRAA/TAAA	TAAA
Aortic coverage	+	++	+++
Limb ischemia time	+	+++	++
Device cost	+/++	+++	+++
Manufacturing time	+	+++	+
Risk of VV-related complications	+	+++	+++
Learning curve	+	+++	++

Low = +, medium = ++, high = +++, SCI = spinal cord ischemia, CM = custom made; VVs = visceral vessels.

**Table 2 jpm-12-01018-t002:** Suggested strategies for spinal cord ischemia protection during F/BEVAR.

Preoperative	Intraoperative	Postoperative
Assessment of spinal collateral network	CSFD and spinal perfusion pressure monitoring	Spinal perfusion pressure monitoring
MIS^2^ACE (technique still under investigation)	Increase in hemoglobin levels and mean arterial pressure	Hemoglobin and arterial pressure monitoring
	MEPs/SSEPs monitoring	Neurologic monitoring
	Systemic hypothermia	MRA/CTA of the spinal cord and rescue CSFD if symptoms arise
	Distal aortic perfusion/early lower limb reperfusion	
	Staging and TASP	
	LSA and IIAs preservation	

CSFD: Cerebrospinal fluid drainage. MEPs: otor evoked potentials. SSEPs: Sensory evoked potentials. TASP: Temporary aneurysm sac perfusion. LSA: Left subclavian artery. IIA: Internal iliac artery. MIS^2^ACE: Minimally invasive segmental artery coil embolization.

## Data Availability

Not applicable.

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
