# Peer review of "Peri-Operative Management of Patients Undergoing Fenestrated-Branched Endovascular Repair for Juxtarenal, Pararenal and Thoracoabdominal Aortic Aneurysms: Preventing, Recognizing and Treating Complications to Improve Clinical Outcomes"

_jpm, 2022, doi:10.3390/jpm12071018_

Round 1

Reviewer 1 Report

I haven't suggestions. It isca great paper. 

Author Response

Thank you very much for the comments!

Reviewer 2 Report

This article is a review of perioperative management in patients undergoing endovascular reconstruction of fenestrated dilated thoracic and abdominal aortic aneurysms.  The authors present minimally invasive treatment of extensive disease involving the abdominal and thoracic aorta. They demonstrate the perioperative management of patients undergoing F/BEVAR for (juxtarenal abdominal aortic aneurysm - JAAA),(pararenal ab-domininal aortic aneurysm - PRAA) or (thoracoabdominal aortic aneurism - TAAA). The authors discuss prevention, diagnosis, and treatment of complications to improve clinical outcomes.  The paper presents the complications and their treatment options of TAAA, TAA, AAA. The article is valuable, interesting provides a partial meta-analysis. 

Author Response

Thank you for the comments.

Reviewer 3 Report

Paper needs some revisions:

- The aim of this review is not clear neither in the abstract nor in the text. Please revise.

- It is not clear how these review was performed, please add a flow diagram.

- Authors should also discuss about new technology as the use of exoscope in surgery, look at recent paper.

Author Response

  • The aim of this review is not clear neither in the abstract nor in the text. Please revise.

Thank you for the comment. In the authors' opinion, the aim of the work was already specified in the abstract as well as in the introduction of the manuscript. From the abstract: "The aim of this manuscript was to provide a narrative review on the peri-operative management of patients undergoing F/BEVAR procedures for juxtarenal abdominal aortic aneurysm (JAAA), pararenal abdominal aortic aneurysm (PRAA) or thoracoabdominal aortic aneurism (TAAA). It will focus on how to prevent, diagnose, and manage the complications ensuing from these complex interventions, in order to improve clinical outcomes." From the introduction: "The aim of this manuscript is to analyze and review, from an experts’ opinion, the perioperative management of patients undergoing F/BEVAR procedures for complex abdominal (JAAA and PRAA) and thoracoabdominal aneurismatic aortic diseases. It will focus on how to prevent, diagnose, and manage the complications ensuing from such complex interventions, in order to improve clinical outcomes." In response to this comment, no changes have been made.

  • It is not clear how these review was performed, please add a flow diagram.

Thank you for the comment. As originally commissioned by the Editors, this work was prepared as a narrative review rather than a systematic review. Narrative reviews are, by their own nature, conducted on an expert opinion basis rather than through a systematic search of the literature, as they aim to provide a broad overview of a certain topic of interest rather than assessing in systematic fashion a much more limited research question. As such, they are not required to provide a PRISMA diagram. In response to this comment, no changes have been made.

  • Authors should also discuss about new technology as the use of exoscope in surgery, look at recent paper.

Thank you for the comment. While we understand the reviewer's point, it should also be noted that advances in the field of vascular surgery happen at a pace that would make most papers outdated in a shot timeframe. As the aim of this review was to focus on perioperative management of patients undergoing complex endovascular aortic repair (rather than analyzing all possible therapeutic alternatives to treat patients with complex aortic disease), adding the required considerations to the manuscript would fall outside its main topic of interest. In response to this comment, no changes have been made.

Reviewer 4 Report

The authors report a detailed summary of the complications and evidence behind a number of techniques intended to reduce complications after complex endografting procedures.  Overall it appears comprehensive and well cited. I only have a few editorial suggestions to help clarify/summarize some of the main points.

- A table comparing the indications and considerations between the different procedures (standard EVAR, FEVAR, BEVAR, F/BEVAR) may be helpful to summarize.

- In discussing the study by Locham et al, rather than describing that all outcomes had p values less than 0.05, I would suggest placing the actual p value for each complication after the comparison of the percentages. Additionally, it is stated that the most frequent complications are endoleaks and TV instability - is there data on the rate of these complications?

- Figures 1, 2, and 6 should include legends to better describe the concept illustrated by the figure. In Figures 3 and 4, the use of arrows or arrowhead would help point at and clarify the points of interest on the CT scan.

- The second sentence of Ischemic vs Hemorrhatic Complications has a period after "intracranial hemorrhage) or" - is the period a typo or is part of the sentence missing?

- To avoid confusion make sure that for all abbreviations that they are defined at the first time it is mentioned. For example, TV comes before target vessels (TV) and BS is not defined as bridging stent.

- Table 1 is a helpful summary of SCI protection, I would consider similar tables for other sections such as acute kidney injury/ischemic complications as well

Author Response

  1. A table comparing the indications and considerations between the different procedures (standard EVAR, FEVAR, BEVAR, F/BEVAR) may be helpful to summarize.

Thank you for your comment. We have added a new table (please see Table 1 in the revised version of the manuscript) in order to reflect this pertinent observation. The following changes have been made to the manuscript: “Indeed, it is well known that achieving a seal in a morphologically hostile aortic neck will make the repair more prone to lower durability over time, while the parallel graft technique is only recommended as bailout alternative. Therefore, fenestrated-branched endografts (whether physician-modified, off-the-shelf, or custom-made commercially manufactured) would represent the best available endovascular treatment option for complex aortic diseases (Table 1).”

  1. In discussing the study by Locham et al, rather than describing that all outcomes had p values less than 0.05, I would suggest placing the actual p value for each complication after the comparison of the percentages. Additionally, it is stated that the most frequent complications are endoleaks and TV instability - is there data on the rate of these complications?

Thank you for your comment. We have rephrased this paragraph by adding the rate of complications (as reported in the study by Locham et al), in order to reflect this pertinent observation. The following changes have been made to the manuscript: “Locham and colleagues published a large study including 16.048 patients who underwent elective EVAR and 1.641 patients who underwent elective FEVAR.  The rates of any type of complication (9.6% vs 11.3%, p= .03), renal injury (4.3% vs 5.8%, p= .004) and neurologic injury (0.4% vs 0.7%, p= .02) were all significantly higher in the FEVAR group compared with the EVAR group.24

  1. Figures 1, 2, 6 and 8 should include legends to better describe the concept illustrated by the figure. In Figures 3 and 4, the use of arrows or arrowhead would help point at and clarify the points of interest on the CT scan.

Thank you for your comment. We have included legends for figures 1, 2, 6 and 8 as well as provided arrows to clarify the points of interest in figures 3 and 4.

  1. The second sentence of Ischemic vs Hemorrhatic Complications has a period after "intracranial hemorrhage) or" - is the period a typo or is part of the sentence missing?

Thank you for the comment. This was only a typo that has been corrected.

  1. To avoid confusion make sure that for all abbreviations that they are defined at the first time it is mentioned. For example, TV comes before target vessels (TV) and BS is not defined as bridging stent.

Thank you for the comment. All abbreviations have been double checked and defined at first time they are mentioned.

  1. Table 1 is a helpful summary of SCI protection, I would consider similar tables for other sections such as acute kidney injury/ischemic complications as well.

Thank you for the comment. Some of these considerations are now included in the new Table 1. Also, we believe the paragraph on acute kidney injury is already quite extensive and can provide the readers with all relevant information. Since the manuscript is already long, we would like to avoid including an additional table. However, we are prepared to do so at a later stage if deemed necessary by the reviewers and editors.

Round 2

Reviewer 3 Report

Good.